

# A comparative analysis of small RNA sequencing data in tubers of purple potato and its red mutant reveals small RNA regulation in anthocyanin biosynthesis

Fang Liu[*], Peng Zhao[*], Guangxia Chen, Yongqiang Wang and Yuanjun Yang

Institute of Vegetables, Shandong Academy of Agricultural Sciences, Jinan, China
[*] These authors contributed equally to this work.

## ABSTRACT

Anthocyanins are a group of natural pigments acting as stress protectants induced by biotic/abiotic stress in plants. Although the metabolic pathway of anthocyanin has been studied in potato, the roles of miRNAs on the metabolic pathway remain unclear. In this study, a purple tetraploid potato of SD92 and its red mutant of SD140 were selected to explore the regulation mechanism of miRNA in anthocyanin biosynthesis. A comparative analysis of small RNAs between SD92 and SD140 revealed that there were 179 differentially expressed miRNAs, including 65 up- and 114 down-regulated miRNAs. Furthermore, 31 differentially expressed miRNAs were predicted to potentially regulate 305 target genes. KEGG pathway enrichment analysis for these target genes showed that plant hormone signal transduction pathway and plant-pathogen interaction pathway were significantly enriched. The correlation analysis of miRNA sequencing data and transcriptome data showed that there were 140 negative regulatory miRNA-mRNA pairs. The miRNAs included miR171 family, miR172 family, miR530b_4 and novel_mir170. The mRNAs encoded transcription factors, hormone response factors and protein kinases. All these results indicated that miRNAs might regulate anthocyanin biosynthesis through transcription factors, hormone response factors and protein kinase.

## INTRODUCTION

Anthocyanins are flavonoid compounds, which are secondary metabolites. They are natural food pigments found in edible parts of fruits, vegetables and crops (*Chiu et al., 2010*). The foods rich in anthocyanin present bright colors and are popular with people (*Bimpilas et al., 2016*). Moreover, anthocyanins also have antioxidant activity and can protect human beings from disease or reduce the damage of disease. The anthocyanin extracts from purple rice protect cardiac function in STZ-induced diabetes rat hearts by inhibiting cardiac hypertrophy and fibrosis (*Chen et al., 2016*). Anthocyanins from red potato show anti-hepatotoxity in rats with toxicity of D-galactosamine (*Han et al., 2006*).

Corresponding author
Yuanjun Yang, yangyuan-jun@263.net.cn

Anthocyanin extracts from bilberries and blackcurrants have protective activity on acute acetaminophen-induced hepatotoxicity in rats (*Cristani et al., 2016*).

In anthocyanin biosynthesis, phenylalanine is a primary precursor. Then under the action of a series of enzymes, the substances of coumaroyl CoA, dihydroflavonols, leucoanthocyanins and anthocyanins are successively produced. Anthocyanin biosynthesis is regulated by structural genes and their transcription factors. Some genes regulating anthocyanin biosynthesis have been isolated and characterized in potato, such as *f3'5'h* (*Jung et al., 2005*), *dfr* (*De Jong et al., 2003*), *developer* (*D*) locus (*Jung et al., 2009*), *AN1* (*D'Amelia et al., 2014*) and *StMYB44* (*Liu et al., 2019*).

Small RNAs usually consist of 20-30 nucleotides and widely exist in eukaryotic organisms. According to their biogenesis modes, small RNAs are distinguished into three major types, namely miRNA, siRNA and piRNA (*Axtell, 2013*; *Chen, 2009*). Small RNAs guide biological processes at DNA or RNA level, for example, the cleavage of complementary RNAs. Different types of small RNAs have similar molecular functions. Both miRNAs and siRNAs can inhibit translation of target mRNAs, and both siRNAs and piRNAs can direct chromatin modifications (*Chen, 2009*). miRNAs regulate target mRNAs through transcript cleavage and/or translational inhibition. Conserved miRNAs play vital roles in many plant physiological processes, such as development, stress responses, primary and secondary metabolism (*Gou et al., 2011*; *Jones-Rhoades, Bartel & Bartel, 2006*; *Matzke et al., 2009*; *Xia et al., 2012*).

So far, miRNAs have been proved to be involved in the regulation of anthocyanin biosynthesis. miRNA858a and HYPOCOTYL 5 (HY5) can repress the expression of *MYB-LIKE 2* (*MYBL2*), thus leading to the activation of anthocyanin biosynthesis pathway (*Wang et al., 2016*). Increasing miR156 activity promotes anthocyanin accumulation, while reducing miR156 activity leads to a high level of flavonol (*Gou et al., 2011*). Both miR828 and miR858 regulate *VvMYB114* to promote anthocyanin biosynthesis in grapes (*Tirumalai et al., 2019*). The miRNA involved in anthocyanin biosynthesis pathway are also reported in apple (*Hu et al., 2021*), tomato (*Jia et al., 2015*), potato (*Bonar et al., 2018*) and kiwifruit (*Li et al., 2019*). However, there are few studies on the post-transcriptional regulation of miRNA in potato anthocyanin biosynthesis. In the study, a comparative miRNA analysis and the expression analysis of miRNA-mRNA were performed between purple flesh potato, SD92, and its red flesh mutant, SD140. These results will shed light on the regulation mechanism of miRNA in potato anthocyanin biosynthesis.

## MATERIALS & METHODS

### Plant materials

SD92, commonly known as Hei Jingang, was a tetraploid potato with purple skin and purple flesh. SD140 is a mutant of SD92. Its skin and flesh colors were red (*Liu et al., 2018*; *Liu et al., 2015*). Two materials were planted in a greenhouse for two months at 20 ± 2 °C with a photoperiod of 16 h light/8 h dark.

## Sample library construction and sequencing

Fresh tubers (diameter 4–5 cm) from three individual plants were harvested for three biological replicates, cleaned with sterilized water, frozen in liquid nitrogen and finally stored at −80 °C. Total RNA extraction of the samples was performed with a modified Trizol reagent (*Liu et al., 2018*) for library construction and validation of miRNA sequencing data.

Small RNA was isolated and the library was constructed in accordance with the protocol of Preparing Samples for Analysis of Small RNA (Illumina, San Diego, CA, USA). The 18-30 nt RNA segments were separated from total RNA by polyacrylamide gel electrophoresis, then ligated with 3′ adaptor (GAACGACATGGCTACGATCCGACTT) and 5′ adaptor (AGTCGGAGGCCAAGCGGTCTTAGGAAGACAA). The resulting segments were employed to synthesize first-strand cDNA. The cDNA was amplified and only cDNA with both 3′ and 5′ adaptors was enriched. Finally, the fragments of 100–120 bp were separated to construct the library. After library quantification and single-stranded DNA cyclization, the library was sequenced by BGISEQ-500 technology. The raw data was deposited into NCBI BioProject database (PRJNA824931).

## miRNA identification and prediction

The impurities of raw data, including 5′ primer contaminants, no-insert tags, oversized insertion tags, low quality tags, poly-A tags and the tags without 3′ primer, were excluded from the raw data to obtain clean tags. Low-quality tags were tags whose base quality values were less than 20, accounting for more than 50% of the total bases. The clean tags were mapped to potato reference genome PGSC_DM v4.03 (http://solanaceae.plantbiology.msu.edu/data) by Bowtie2 (*Langmead et al., 2009*) and small RNA databases miRBase (*Kozomara & Griffiths-Jones, 2014*), snoRNA (*Yoshihama, Nakao & Kenmochi, 2013*) and Rfam (*Nawrocki et al., 2015*). If a small RNA could be mapped to more than one database, the small RNA annotation followed the searching priority of miRBase > snoRNA > Rfam. One small RNA was only mapped to one category. The small RNAs mapped to Rfam database were validated by cmsearch (*Nawrocki & Eddy, 2013*). The novel miRNA was determined by miRA (*Evers et al., 2015*) according to the characteristic hairpin structure of miRNA precursor. Small interfering RNA (siRNA), a 22–24 nt double-strand RNA, was identified by the characteristic of one strand 2 nt shorter than the other (*Jagla et al., 2005*).

## miRNA expression and screening of differentially expressed miRNAs (DEMs)

The expression level of miRNA was estimated by the transcripts per kilobase million (TPM) (*'t Hoen et al., 2008*). The differential expression was calculated by DEGseq (*Wang et al., 2010*) based on MA-plot method (*Yang et al., 2002*). The $P$-value calculated for each gene was adjusted to Q-value for multiple testing corrections by two alternative strategies. The miRNAs with expression fold change > 2 and Q-value < 0.001 were defined as differentially expressed miRNAs. The volcano plot and heatmap of differentially expressed miRNAs were obtained by Excel 2016 and MeV (*Saeed et al., 2003*), respectively.

**Table 1  Primer sequences of miRNAs for real-time quantitative PCR.**

| Primer | Direction | Sequence (5′–3′) |
|---|---|---|
| 18S rRNA | Forward | CCTGGTCGGCATCGTTTA |
| 18S rRNA | Reverse | CGAACAACTGCGAAAGCAT |
| miR156a-5p | Forward | TGACAGAAGAGAGTGAGCAC |
| miR166a-3p | Forward | TCGGACCAGGCTTCATTCC |
| miR166d-5p_2 | Forward | GGAATGTTGTCTGGCTCGAGG |
| miR171b-3p | Forward | TTGAGCCGTGCCAATATCAC |
| miR171b-3p_2 | Forward | TTGAGCCGCGTCAATATCTCT |
| miR172b | Forward | GGAATCTTGATGATGCTGCA |
| miR172e-5p | Forward | GCAACATCATCAAGATTCACA |
| miR399a_6 | Forward | GCCAAAGGAGAATTGCCCTG |
| miR399i | Forward | CCAAAGGAGAGCTGCCCTG |
| miR399j_2 | Forward | TGCCAAAGGAGAGTTGCCCTA |
| miR530a | Forward | TGCATTTGCACCTGCACCTT |
| miR828a_1 | Forward | CGCTGTCTTGCTCAAATGAGTATTC |
| novel_mir32 | Forward | ATTAACTTTGGCCAGCATC |
| novel_mir105 | Forward | GGACCCTTGGCGAAGTCACC |
| novel_mir143 | Forward | CACTGAGTTGGACCCTTGGC |
| novel_mir170 | Forward | GCGAGCGAATTAGATTCATTGTTTGA |

## Target gene prediction, Gene Ontology (GO) and KEGG pathway enrichment analyses

TargetFinder (*Fahlgren & Carrington, 2010*) and psRobot (*Wu et al., 2012*) were used to predict the target genes of miRNAs. All target genes were mapped to GO-terms in the database (http://www.geneontology.org/) and KEGG Orthology (*Kanehisa et al., 2008*) pathways. The enrichment analyses of GO terms and KEGG pathways were performed by the hypergeometric test based on GO::TermFinder (*Boyle et al., 2004*). The P-values were adjusted by Bonferroni method (*Abdi, 2007*). The adjusted P-value was defined as Q-value. The terms with Q-value < 0.05 were defined as significantly enriched terms.

## Expression validation of miRNAs

RNAs were digested by DNaseI (Thermo, USA) to remove genome DNA. First-strand cDNA was synthesized by miRNA First Strand cDNA Synthesis Kit (Sangon Biotech, China) using tailing reaction method. Real-time quantitative PCR (RT-qPCR) was performed with UltraSYBR Mixture Kit (CWBIO, China) by using *18S rRNA* (GenBank: X67238.1) as a reference gene. The primers of *18S rRNA* and miRNAs were listed in Table 1. The universal reverse primer for miRNAs was supplied from miRNA First Strand cDNA Synthesis Kit. Three biological replicates were performed. Significant difference of miRNA expression between SD92 and SD140 was identified by Student's $t$-test ($P < 0.05$).

**Table 2  Summary of sequencing data for each sample.**

| Sample name | Raw tag count | Low quality tag | Invalid adapter tag | Poly A tag | Tag length < 18 | Clean tag | Q20 of clean tag (%) | Percentage of clean tag (%) |
|---|---|---|---|---|---|---|---|---|
| SD140_1 | 30,152,601 | 521,573 | 1,211,217 | 765 | 296,890 | 28,122,156 | 99.30 | 93.27 |
| SD140_2 | 29,662,224 | 559,145 | 642,637 | 1,307 | 285,077 | 28,174,058 | 99.20 | 94.98 |
| SD140_3 | 29,108,569 | 439,201 | 1,438,318 | 979 | 420,200 | 26,809,871 | 99.20 | 92.10 |
| SD92_1 | 28,058,311 | 476,281 | 601,154 | 814 | 262,128 | 26,717,934 | 99.00 | 95.22 |
| SD92_2 | 28,907,701 | 462,036 | 684,333 | 2,174 | 265,810 | 27,493,348 | 99.30 | 95.11 |
| SD92_3 | 29,706,600 | 544,647 | 816,486 | 1,600 | 341,405 | 28,002,462 | 99.20 | 94.26 |

## RESULTS

### Sequencing and classification of potato small RNAs

To identify the miRNAs regulating potato flesh color, six small RNA libraries were constructed and sequenced. The counts of raw tags of six libraries ranged from 28,058,311 to 30,152,601 (Table 2). Low quality tags, invalid adapter tags, poly-A tags and short valid length tags (shorter than 18 nt) were removed to obtain clean tags. The percentages of clean tags of six libraries ranged from 92.10% to 95.22%, which indicated the sequencing data could be used for subsequent analyses. Of the six libraries, 19-25 nt length tags accounted for 87.9%–96.4% of the total tags, and the 24 nt tags were the most abundant (Table S1). More than 85.04% of the total clean tags from six libraries were mapped to the reference genome (Table S2). Therefore, the sequencing data should accurately reflect small RNA expression and could be used for differential expression analysis of small RNA. To classify and annotate small RNAs, the clean tags were mapped to small RNA databases miRBase, snoRNA and Rfam. The types and proportion of identified small RNAs were similar within six libraries. The intergenic miRNAs were the most abundant (Table S3).

### Identification of known and novel miRNAs

There were about 300 known miRNAs and 160 novel miRNAs identified in every library (Table 3). In total, 356 known miRNAs belonging to 121 miRNA families were identified (Table S4), and miR172 family was the most abundant family where 21 members were identified. The nucleotide bias analyses on these non-redundant known miRNAs (Fig. S1A) showed that the first and 24th nucleotide preferred to be uracil (U), and adenine (A) was the dominant nucleotide in the 10th nucleotide position. Meanwhile, several nucleotide positions were conserved. For example, the proportions of four kinds of nucleotides were nearly equal in the 4th, 9th and 16th nucleotide position (Fig. S1A).

Unmapped tags were further used to predict novel small RNAs. Totally, 171 novel miRNAs were identified for six libraries. The mature sequences, star sequences and precursor sequences of 171 novel miRNAs were listed in Table S5. The length of the novel miRNAs ranged from 19 to 30 nucleotides. Most of the nucleotide positions preferred to be uracil (U) or adenine (A) (Fig. S1B). Two exceptions were the 9th and 11th nucleotide where the dominant nucleotides were guanine (G) and cytosine (C), respectively.

**Table 3  Summary of detected small RNAs for each sample.**

| Sample name | Known miRNA | Novel miRNA | Known siRNA | Novel siRNA |
|---|---|---|---|---|
| SD140_1 | 290 | 151 | 0 | 12,518 |
| SD140_2 | 293 | 161 | 0 | 13,671 |
| SD140_3 | 284 | 145 | 0 | 12,447 |
| SD92_1 | 275 | 166 | 0 | 13,373 |
| SD92_2 | 304 | 161 | 0 | 11,225 |
| SD92_3 | 311 | 168 | 0 | 13,147 |

## Differentially expressed miRNAs between SD92 and SD140

To further validate the expression changes of miRNAs between SD92 and SD140, 15 miRNAs from 11 different miRNA families were randomly selected to be tested by RT-qPCR (Fig. 1). The results of RT-qPCR showed the same expression regulation pattern with miRNA sequencing data, which suggested that the miRNA sequencing result was reliable. What's more, the results showed 6 miRNAs were differentially expressed between SD92 and SD140 ($P < 0.05$). Different miRNAs from the same miRNA family displayed the same regulation pattern. For example, both miR166a-3p and miR166d-5p_2 were from miR166 family and exhibited higher expression levels in SD140 than in SD92.

A total of 179 differentially expressed miRNAs were identified in this study, including 107 known miRNAs and 72 novel miRNAs (Fig. 2A, Table S6). Among the differentially expressed miRNAs, 65 and 114 were confirmed to be up- and down-regulated in SD140, respectively. These miRNAs belonged to 49 miRNA families. Of the 49 miRNA families, miR399 and miR172 family were the two largest families, which contained 10 and 9 miRNA members, respectively. Interestingly, the members of miR399 and miR172 families were significantly down-regulated in SD140, respectively.

## Target gene prediction of miRNAs

To further explore the function of miRNAs, the target genes (mRNAs) of all miRNAs were predicted by psRobot and TargetFinder. Totally, 7,416 target genes were identified for 450 miRNAs where 897 target genes were confirmed as targets of 116 miRNAs by both softwares. Among these 897 target genes, 305 genes were regulated by 31 differentially expressed miRNAs (Table S7).

## GO and KEGG pathway enrichment analysis of target genes

GO enrichment analysis of the above 305 target genes showed that the biological process ontology included 47 GO terms. "Cellular macromolecule metabolic process" and "macromolecule metabolic process" were the most abundant GO terms, containing 77 genes, respectively.

The cellular component ontology included 16 GO terms, and the most abundant terms were "cell" and "cell part", which contained 115 genes, respectively. The molecular

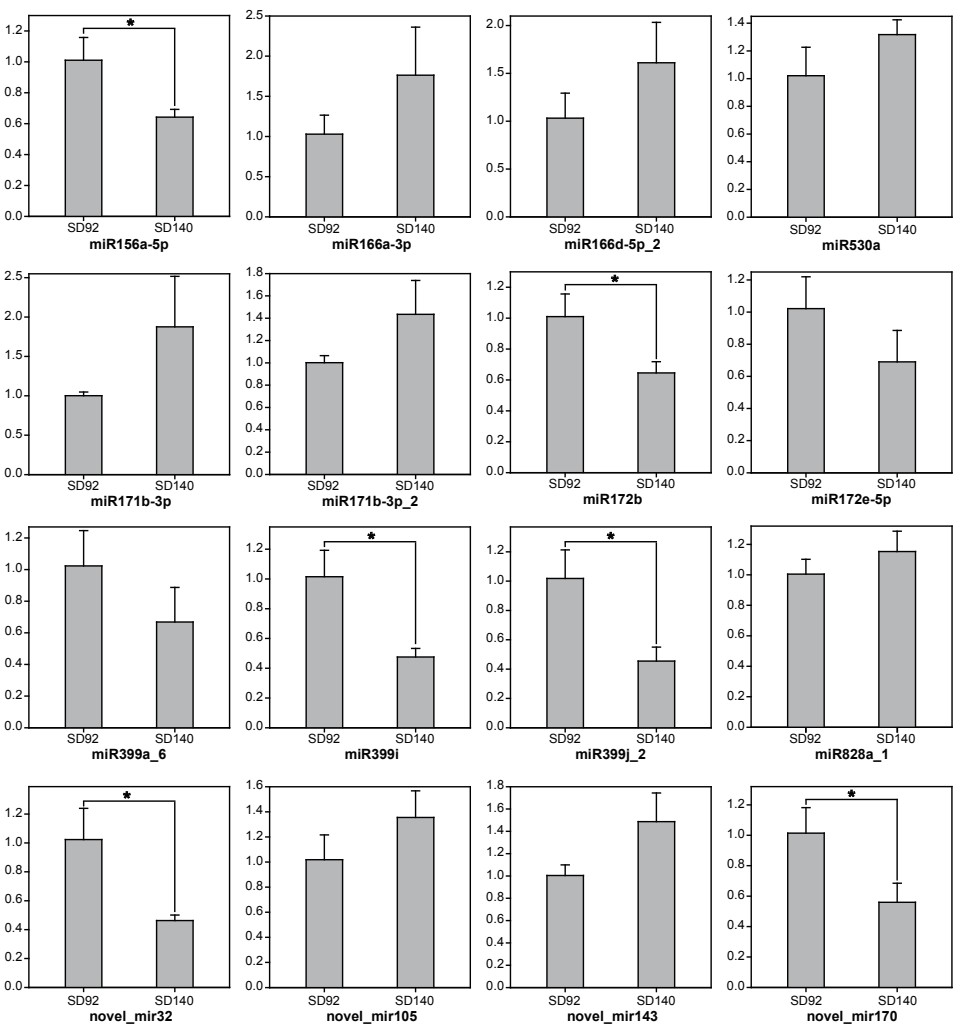

**Figure 1 Expression analysis of miRNAs by RT-qPCR.** The values are represented by mean ± standard deviation ($n = 3$). Student's $t$-test, $P < 0.05$.

function ontology included 10 GO terms. The GO term "binding" contained 126 genes, which was the most abundant term in molecular function (Fig. 3).

To explore the possible function of target genes, KEGG pathway enrichment analysis was performed. The 305 target genes of 31 DEMs were distributed in 6 first-level and 33 second-level KEGG pathways, respectively. The first-level KEGG pathway term "metabolism" was the most abundant, including 10 second-level KEGG pathway terms. Thirty-eight target genes were assigned in the second-level KEGG pathway term "signal transduction", which was the most abundant second-level KEGG pathway term (Fig. 4).

Among the enriched top 20 pathways, only two pathways, "plant hormone signal transduction" and "plant-pathogen interaction", were defined as significantly enriched pathways ($P < 0.05$), which comprised 24 target genes, respectively (Fig. 5 and Table

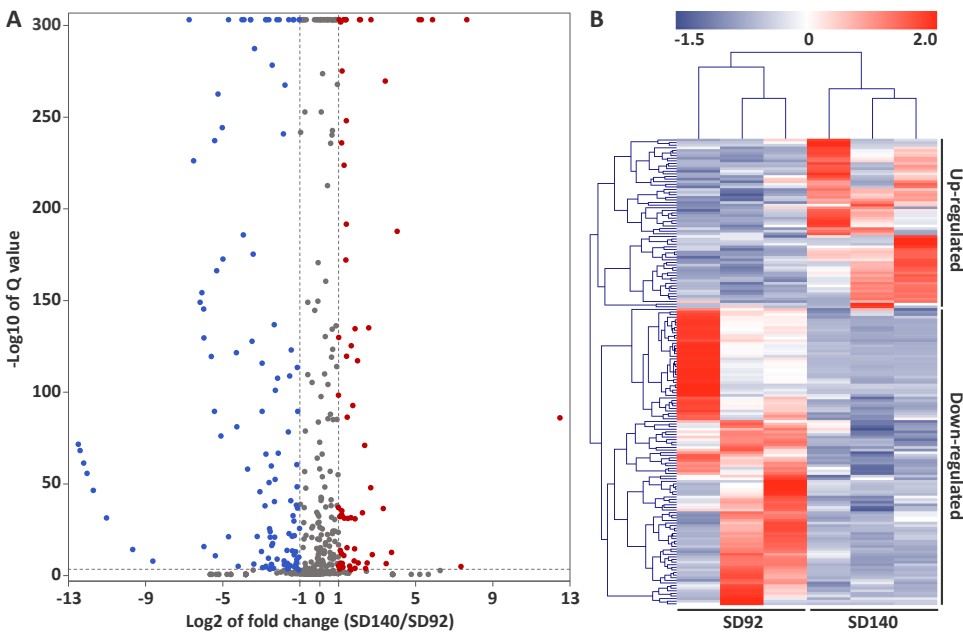

**Figure 2** **Identification of differentially expressed miRNAs between SD92 and SD140.** (A) Volcano plot of differentially expressed miRNAs between SD92 and SD140. The cutoff values of fold change and *Q*-value are > 2 and < 0.001, respectively. Up-regulated and down-regulated miRNAs are indicated by red and blue dots. (B) Heatmap of differentially expressed miRNAs in three biological replicates. Hierarchical clustering was performed by complete linkage method and Euclidean distance.

S8). This indicated that the DEMs between SD92 and SD140 might be involved in plant-pathogen interaction and hormone signal transduction.

## Target genes of miRNAs involved in regulation of anthocyanin biosynthesis

Generally, plant miRNAs regulate target mRNAs through two major mechanisms, transcript cleavage and translational inhibition (*Chen, 2009*), thus there are negative regulation relationship in the expressions of miRNA and corresponding target genes. In our previous study, a comparative transcriptome analysis was performed between SD92 and SD140 (*Liu et al., 2018*). By combining transcriptome sequencing data (SRA accession number: SRP125987) and miRNA sequencing data of present study, 31 differentially expressed miRNAs and corresponding target mRNAs were identified and listed in Table S9. Among them, the differentially expressed miRNAs negatively regulating target mRNAs were screened, and 140 miRNA-mRNA pairs were confirmed. In these miRNAs-mRNAs pairs, miRNAs contained 5 known miRNA families and 12 novel miRNAs. These mRNAs corresponded to 71 genes (Table 4). These genes mainly encoded transcription factors, quamosa promoter binding protein, hormone response factors, protein kinases and disease resistance protein.

Transcription factors affect anthocyanin biosynthesis by regulating the expressions of structural genes (*D'Amelia et al., 2014*; *Liu et al., 2016*). In this study, we focused on the regulation of miRNA on transcription factors in anthocyanin biosynthesis (Table 4).

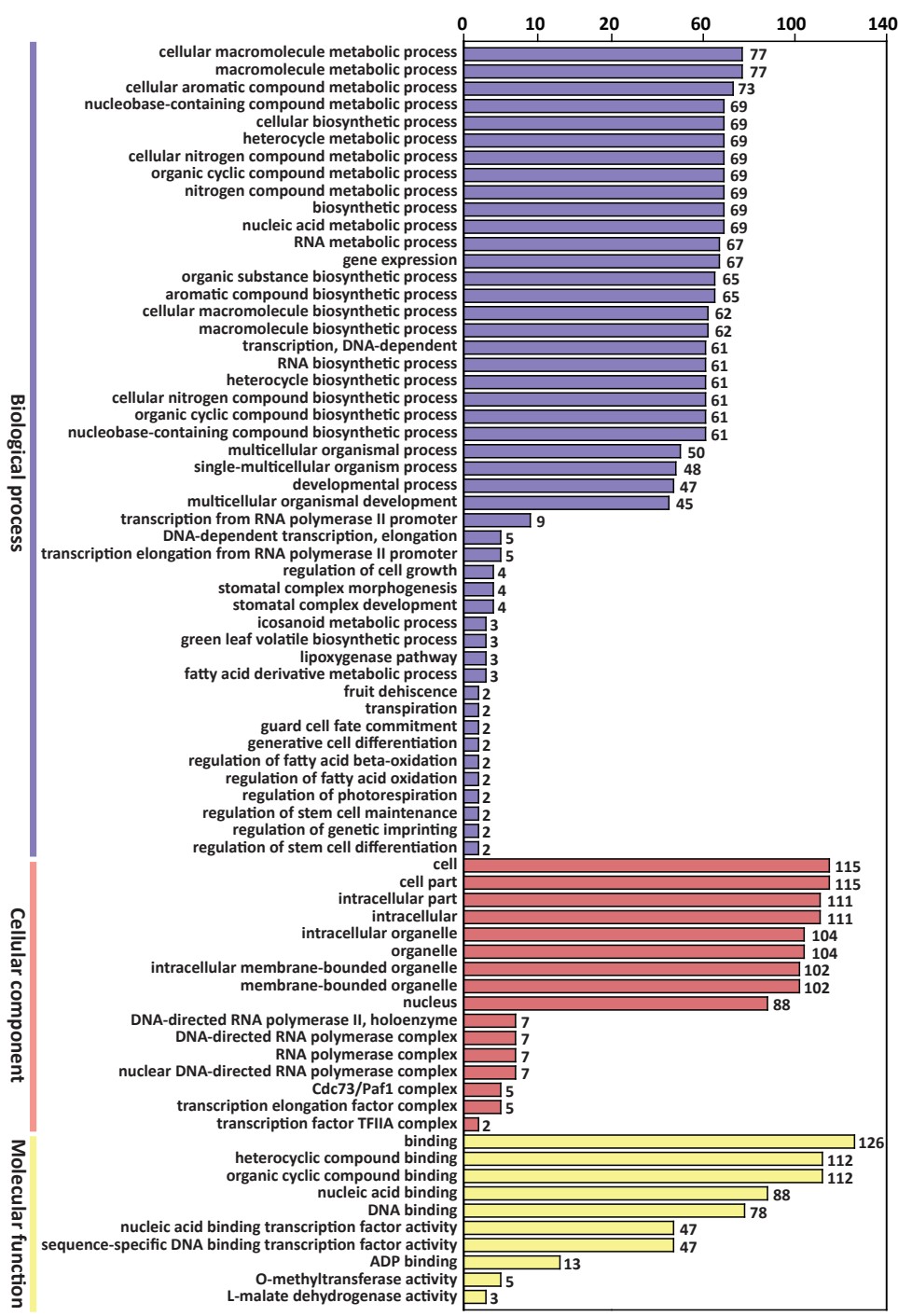

**Figure 3  GO classification of predicted target genes of the differentially expressed miRNAs.**

*PGSC0003DMG400006604*, *PGSC0003DMG400011046* and *PGSC0003DMG400012038*, which were regulated by miR172b, encoded AP2 transcription factor SlAP2e, RAP2-7-like and RAP2-7, respectively. The target gene of miR530b_4, *PGSC0003DMG400025479*,

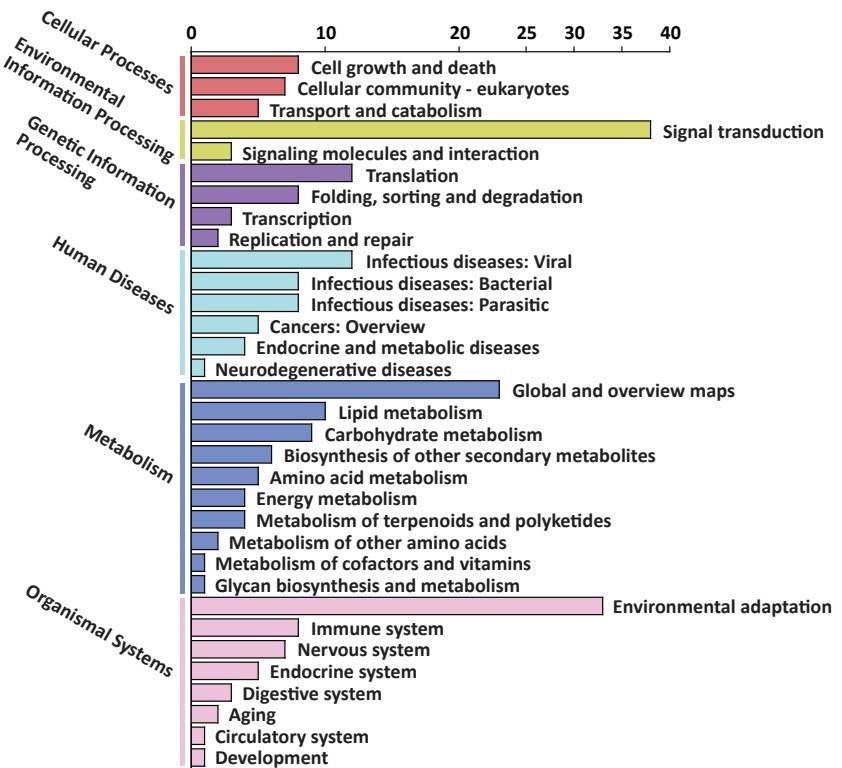

**Figure 4   First-level and second-level KEGG pathway classification of predicted target genes of the DEMs.** Six different first-level KEGG pathway are distinguished in different colors.

encoded AP2-like transcription factor TOE3. *PGSC0003DMG400011457* encoded WRKY transcription factor 48 and was regulated by miR172e-5p. Both *PGSC0003DMG400004826* and *PGSC0003DMG400018279*, which were regulated by novel_mir170, encoded transcription factor ERF039-like and MYB35-like, respectively.

Hormones improve the biosynthesis of anthocyanins (*Zhang et al., 2011*; *Palma-Silva et al., 2016*), so we did research on miRNA regulating hormones in this experiment in order to throw light on miRNA regulation mechanism on anthocyanins biosynthesis. In this study, RAP2-7 and RAP2-7-like, which were regulated by miR172b, were ethylene-responsive transcription factors. TOE3 transcription factor, which was regulated by miR172b and miR530b_4, was also responsive to ethylene (Table 4). The target gene of miR171b-3p, *PGSC0003DMG400012683*, encoded the DELLA protein that was an inhibitor of GA signal transduction.

Protein kinases are involved in anthocyanin biosynthesis (*Li et al., 2016*). Protein kinases regulated by miRNA were investigated in this study. Both *PGSC0003DMG400018811* and *PGSC0003DMG400024795*, which were regulated by novel_mir170, encoded LRR receptor-like serine/threonine protein kinase ERECTA and RCH1, respectively. *PGSC0003DMG400026383* encoded receptor-like protein kinase and was regulated by novel_mir117.

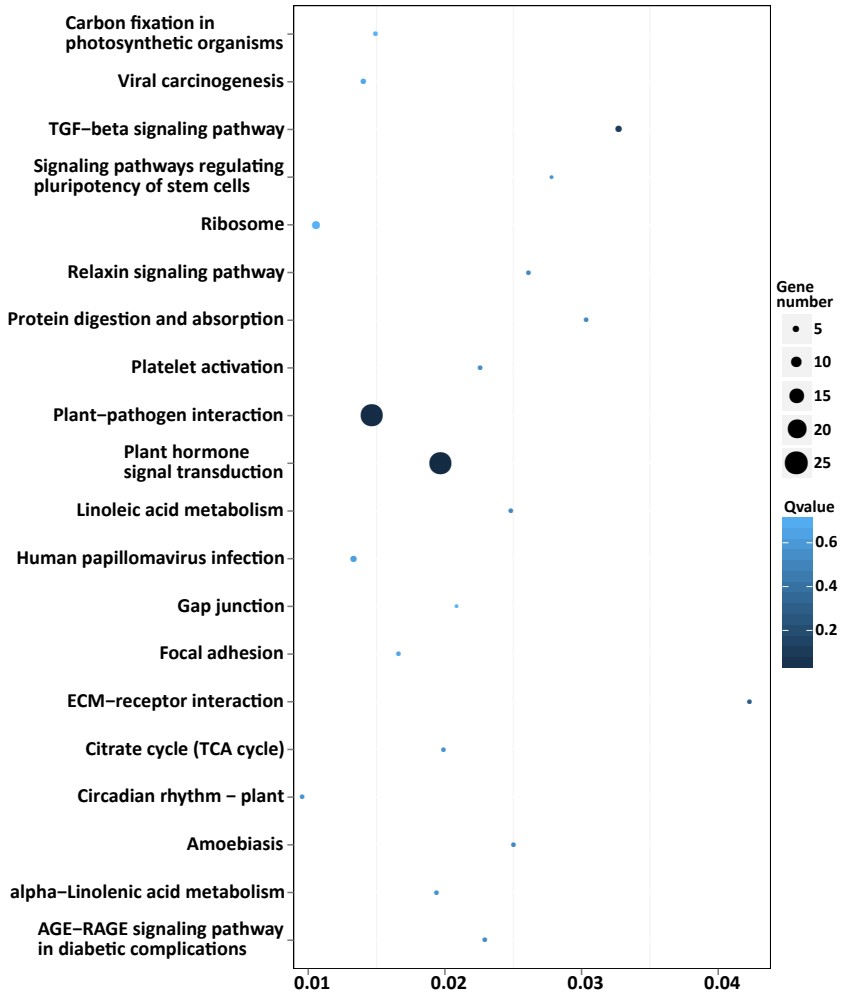

**Figure 5**  **Scatterplot of enriched KEGG pathways of predicted target genes of the DEMs.** *X* axis indicates the rich factor. The rich factor is the ratio of DEMs target gene numbers annotated in the pathway term to all gene numbers annotated in the pathway. *Y* axis indicates KEGG pathways.

There were also significant changes in the expression of target genes regulated by other miRNAs, such as *PGSC0003DMG402007414*, which was target gene of novel_mir105 and novel_mir143, but the gene function was unknown.

## DISCUSSION

Generally, miRNAs play an important role in some kinds of plant biological processes such as growth, development and stress response (*Jones-Rhoades, Bartel & Bartel, 2006*). The functions of miRNAs in plant anthocyanin biosynthesis have been reported in some species, including Arabidopsis (*Gou et al., 2011*; *Wang et al., 2016*), apple (*Hu et al., 2021*), grape (*Tirumalai et al., 2019*), tomato (*Jia et al., 2015*), sweet potato (*He et al., 2019*) and kiwi fruit (*Li et al., 2019*).

**Table 4  Differentially expressed miRNAs and negatively regulated target genes.**

| miRNA | Target gene | Gene annotation |
| --- | --- | --- |
| miR156a-5p | *PGSC0003DMG400022824* | Squamosa promoter-binding protein 1-like |
| miR156a-5p | *PGSC0003DMG400023962* | Uncharacterized protein |
| miR156a-5p | *PGSC0003DMG400029156* | Cell cycle checkpoint protein RAD17 |
| miR156a-5p | *PGSC0003DMG400032817* | Squamosa promoter-binding protein 1-like |
| miR156a-5p | *PGSC0003DMG400034310* | Squamosa promoter-binding-like protein 12 |
| miR171b-3p | *PGSC0003DMG400009015* | DEAD-box ATP-dependent RNA helicase 24 |
| miR171b-3p | *PGSC0003DMG400012683* | DELLA protein |
| miR172b | *PGSC0003DMG400004006* | Floral homeotic protein APETALA 2 |
| miR172b | *PGSC0003DMG400006604* | AP2 transcription factor SlAP2e |
| miR172b | *PGSC0003DMG400011046* | Ethylene-responsive transcription factor RAP2-7-like |
| miR172b | *PGSC0003DMG400012038* | Ethylene-responsive transcription factor RAP2-7 |
| miR172b | *PGSC0003DMG400027904* | Floral homeotic protein APETALA 2-like |
| miR172b | *PGSC0003DMG400030080* | Phosphatidylinositol/phosphatidylcholine transfer protein SFH4 |
| miR172b & miR530b_4 | *PGSC0003DMG400025479* | AP2-like ethylene-responsive transcription factor TOE3 |
| miR172e-5p | *PGSC0003DMG400010386* | Malate dehydrogenase, glyoxysomal |
| miR172e-5p | *PGSC0003DMG400011457* | Probable WRKY transcription factor 48 |
| miR172e-5p | *PGSC0003DMG400011477* | Putative lysine-specific demethylase JMJ16 |
| miR172e-5p | *PGSC0003DMG400021020* | Uncharacterized protein |
| miR172e-5p & novel_mir32 | *PGSC0003DMG400014214* | Uncharacterized protein |
| miR482e-5p & novel_mir117 | *PGSC0003DMG400030780* | Uncharacterized protein |
| miR530a | *PGSC0003DMG400010027* | Dof zinc finger protein DOF3.5-like |
| miR530a | *PGSC0003DMG400022193* | Pirin-like protein |
| miR530a | *PGSC0003DMG400030421* | Transcription initiation factor IIA large subunit |
| miR530a | *PGSC0003DMG400038860* | Uncharacterized protein |
| miR530b_4 | *PGSC0003DMG400001126* | Uncharacterized protein |
| miR530b_4 | *PGSC0003DMG400030587* | Non-specific lipid-transfer protein 2-like |
| novel_mir32 | *PGSC0003DMG400003436* | Uncharacterized protein |
| novel_mir32 | *PGSC0003DMG400007187* | Probable protein S-acyltransferase 1 |
| novel_mir32 | *PGSC0003DMG400009055* | Uncharacterized protein |
| novel_mir32 | *PGSC0003DMG400011113* | Putative disease resistance protein RGA3 |
| novel_mir32 | *PGSC0003DMG400012875* | Protein disulfide isomerase-like 1-3 |
| novel_mir32 | *PGSC0003DMG400016798* | Polyadenylate-binding protein 2-like |
| novel_mir32 | *PGSC0003DMG400017569* | Protein disulfide-isomerase-like |
| novel_mir32 | *PGSC0003DMG400027301* | Caffeic acid 3-O-methyltransferase-like |
| novel_mir32 | *PGSC0003DMG400032155* | Linoleate 13S-lipoxygenase 2-1, chloroplastic |
| novel_mir32 | *PGSC0003DMG400043688* | Uncharacterized protein |
| novel_mir42 | *PGSC0003DMG400008897* | L-type lectin-domain containing receptor kinase IV.1-like |
| novel_mir54 | *PGSC0003DMG400032120* | UPF0496 protein At3g19330-like |
| novel_mir61 | *PGSC0003DMG400004296* | Late blight resistance protein homolog R1B-16 |

**Table 4** (*continued*)

| miRNA | Target gene | Gene annotation |
| --- | --- | --- |
| novel_mir61 | *PGSC0003DMG400004756* | Late blight resistance protein homolog R1A-10 |
| novel_mir61 | *PGSC0003DMG400007867* | Disease resistance protein RGH3 |
| novel_mir61 | *PGSC0003DMG400007870* | Late blight resistance protein homolog R1A-3 |
| novel_mir61 | *PGSC0003DMG400007872* | Late blight resistance protein homolog R1C-3 |
| novel_mir61 | *PGSC0003DMG400031244* | THUMP domain-containing protein 1 homolog |
| novel_mir61 | *PGSC0003DMG402007871* | Disease resistance protein RGH3 |
| novel_mir67 | *PGSC0003DMG400008560* | Uncharacterized protein |
| novel_mir67 | *PGSC0003DMG400017053* | Uncharacterized protein |
| novel_mir67 | *PGSC0003DMG400030551* | Multicopper oxidase LPR2 |
| novel_mir75 | *PGSC0003DMG400003887* | Uncharacterized protein |
| novel_mir75 | *PGSC0003DMG400009731* | Probable S-adenosylmethionine-dependent methyltransferase |
| novel_mir75 | *PGSC0003DMG400017312* | RING finger protein 44 |
| novel_mir75 | *PGSC0003DMG400025978* | Uncharacterized protein |
| novel_mir78 | *PGSC0003DMG400000774* | RNA-binding protein 2 |
| novel_mir89 | *PGSC0003DMG400006945* | Senescence-associated carboxylesterase 101-like |
| novel_mir105 & novel_mir143 | *PGSC0003DMG402007414* | Uncharacterized protein |
| novel_mir117 | *PGSC0003DMG400020645* | ycf54-like protein |
| novel_mir117 | *PGSC0003DMG400026383* | Probable receptor-like protein kinase |
| novel_mir117 | *PGSC0003DMG400031180* | Uncharacterized protein |
| novel_mir128 | *PGSC0003DMG400034633* | Uncharacterized protein |
| novel_mir128 | *PGSC0003DMG400037457* | Uncharacterized protein |
| novel_mir128 | *PGSC0003DMG400043850* | Uncharacterized protein |
| novel_mir170 | *PGSC0003DMG400000513* | Galactinol-sucrose galactosyltransferase 5 |
| novel_mir170 | *PGSC0003DMG400002541* | 60S ribosomal protein L37-3 |
| novel_mir170 | *PGSC0003DMG400004826* | Ethylene-responsive transcription factor ERF039-like |
| novel_mir170 | *PGSC0003DMG400007189* | Proteasome subunit alpha type-3-like, partial |
| novel_mir170 | *PGSC0003DMG400008432* | Uncharacterized protein |
| novel_mir170 | *PGSC0003DMG400012159* | KAT8 regulatory NSL complex subunit 3 |
| novel_mir170 | *PGSC0003DMG400018279* | Transcription factor MYB35-like |
| novel_mir170 | *PGSC0003DMG400018811* | LRR receptor-like serine/threonine-protein kinase ERECTA |
| novel_mir170 | *PGSC0003DMG400024795* | LRR receptor-like serine/threonine-protein kinase RCH1 |
| novel_mir170 | *PGSC0003DMG400033933* | Hypothetical protein SDM1_41t00024 |

In this study, miR399 and miR172 families were the two largest differentially expressed miRNA families. The expressions of miR399 family (miR399a_6, miR399i, miR399j_2) and miR172 family (miR172e-5p, miR172b) were down-regulated in SD140. miR172 inhibits flavonoid biosynthesis through suppressing the expression of an AP2 transcription factor that positively regulates *MdMYB10* (*Ding et al., 2022*). In SD140, miR172b was down-regulated, and its target gene encoding AP2-like factor was up-regulated, indicating that miR172b regulated the change in anthocyanin biosynthesis from petunidin to pelargonidin through AP2-like factor. Both miR399 expression and anthocyanin accumulation are increased under Pi-deficiency conditions (*Chen et al., 2018*; *Hsieh et al., 2009*). miR399 is related to anthocyanin accumulation. However, the target gene of miR399 was unknown

in SD92 and SD140, so the regulation mechanism of miR399 in anthocyanin biosynthesis remains unclear and needs further study.

miR171 family (miR171a-3p, miR171b-3p, miR171b-3p_2) was up-regulated in SD140 (Table S6). miR171 is down-regulated and anthocyanin accumulation is up-regulated under water deficit (*Ghorecha et al., 2014*). miR171 is related with anthocyanin accumulation. The target gene of miR171b-3p, *PGSC0003DMG400012683*, encoded DELLA protein. DELLA proteins are important repressors of GA signaling (*Chai et al., 2022*; *Sukiran et al., 2022*). Plant hormones are involved in anthocyanin biosynthesis, such as auxin (*Ji et al., 2015*; *Liu, Shi & Xie, 2014*), abscisic acid (ABA) (*Balint & Reynolds, 2013*; *Leão et al., 2014*) and gibberellic acid (GA) (*Loreti et al., 2008*). GA represses the sucrose accumulation in anthocyanin synthesis (*Loreti et al., 2008*) and decreases anthocyanin accumulation under low temperature or phosphate starvation (*Jiang et al., 2007*; *Zhang et al., 2011*). Moreover, the KEGG pathway "plant hormone signal transduction" comprising of 24 target genes was significantly enriched in this study, which suggested that plant hormones were involved in the anthocyanin biosynthesis in SD92 and SD140. Thus, it indicated that miR171b-3p probably regulated the change of anthocyanin biosynthesis in SD92 and SD140 through DELLA protein.

miR828 are frequently reported to be involved in anthocyanin biosynthesis regulation (*Bonar et al., 2018*; *Tirumalai et al., 2019*). In potato, miR828 is associated with purple tuber skin and flesh color rich in anthocyanin. One member of miR828 family, miR828a_1, was identified in SD92 and SD140, but was not significantly expressed differentially between SD92 and SD140. These results indicated that miR828a_1 might not regulate the change of anthocyanin biosynthesis between SD92 and SD140.

The accumulation of anthocyanin is reported to be related with miR156 (*Gou et al., 2011*). In this study, miRNA156 was differentially expressed between SD92 and SD140. Its target gene mainly encoded squamosa promoter binding protein and cell cycle checkpoint protein RAD17. These target genes regulated by miR156a-5p need further study in anthocyanin biosynthesis.

A novel miRNA, novel_mir170, was down-regulated in SD140 (4.81 *vs* 0.14). It regulated a number of target genes, which mainly encoded protein kinase, ethylene responsive transcription factor ERF039-like and transcription factor MYB35-like. Protein kinases play an important role in anthocyanin biosynthesis. Plant sucrose-nonfermenting 1 (SNF1)-related protein kinase is involved in anthocyanin accumulation regulated by MdbHLH3 (*Liu et al., 2017*; *Shen et al., 2017*). Anthocyanin biosynthesis is regulated by mitogen-activated protein kinase (*Luo et al., 2017*; *Wersch, Gao & Zhang, 2018*). In this experiment, the two target genes of novel_mir170 encoding LRR receptor-like serine/threonine-protein kinase were up-regulated, which were consistent with the metabolism data (*Liu et al., 2022*). These results showed that novel_mir170 regulated the change of anthocyanin biosynthesis through LRR receptor-like serine/threonine-protein kinase in SD92 and SD140. MYB transcription factor can regulate the biosynthesis of anthocyanin by regulating the expression of structural genes (*D'Amelia et al., 2014*). The target gene of novel_mir170, which encoded MYB transcription factor, was up-regulated. These results showed that novel_mir170 regulated the anthocyanin biosynthesis by regulating the expression of MYB. Ethylene is closely

related to the biosynthesis of anthocyanin (*Chen et al., 2022*; *Jeong et al., 2010*). In this study, the target gene of novel_mir170 encoding ethylene responsive transcription factor ERF039 was up-regulated. These results indicated that novel_mir170 regulated anthocyanin biosynthesis by up-regulating the expression of ethylene responsive transcription factor. In conclusion, novel_mir170 was an important novel miRNA identified in this study and might be an important miRNA for regulation of anthocyanin biosynthesis.

## CONCLUSIONS

A comparative small RNA sequencing analysis between purple potato and its mutant revealed that there were 179 differentially expressed miRNAs, consisting of 65 up- and 114 down-regulated miRNAs, respectively. miR399 and miR172 families were the two largest differentially expressed miRNA families. A total of 31 differentially expressed miRNAs were predicted to potentially regulate 305 target genes. The miRNA sequencing data and the transcriptome data showed that miR171 family and miR172 family regulated the change in anthocyanin biosynthesis from petunidin to pelargonidin through DELLA protein and AP2-like transcription factor, respectively. A novel miRNA, novel_mir170, regulated anthocyanin biosynthesis by serine/threonine-protein kinase and MYB transcription factor.

## ACKNOWLEDGEMENTS

We sincerely appreciate Dr. Yumeng Huo for his help in designing miRNA primers for RT-qPCR.

### Funding

This work was supported by the National Natural Science Foundation of China (Grant No. 31901589), the Modern Agriculture Industrial Technology System Funding of Shandong Province (Grant No. SDAIT-16-05), and the ''333'' Project of Shandong Academy of Agricultural Sciences-Molecular Breeding of Vegetables and Flowers (Grant No. CXGC2021B17). The funders had no role in study design, data collection and analysis, decision to publish, or preparation of the manuscript.

### Grant Disclosures

The following grant information was disclosed by the authors:
National Natural Science Foundation of China: 31901589.
Modern Agriculture Industrial Technology System Funding of Shandong Province: SDAIT-16-05.
''333'' Project of Shandong Academy of Agricultural Sciences-Molecular Breeding of Vegetables and Flowers: CXGC2021B17.

### Competing Interests

The authors declare there are no competing interests.

## Author Contributions

- Fang Liu performed the experiments, analyzed the data, prepared figures and/or tables, and approved the final draft.
- Peng Zhao performed the experiments, analyzed the data, authored or reviewed drafts of the article, and approved the final draft.
- Guangxia Chen performed the experiments, prepared figures and/or tables, and approved the final draft.
- Yongqiang Wang performed the experiments, authored or reviewed drafts of the article, and approved the final draft.
- Yuanjun Yang conceived and designed the experiments, authored or reviewed drafts of the article, and approved the final draft.

## Data Availability

The data is available at NCBI: PRJNA824931.

## Supplemental Information

Supplemental information for this article can be found online at http://dx.doi.org/10.7717/peerj.15349#supplemental-information.

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
