# Peer review of "A comparative analysis of small RNA sequencing data in tubers of purple potato and its red mutant reveals small RNA regulation in anthocyanin biosynthesis"

_PeerJ, doi:10.7717/peerj.15349_

## Round 0.1 · original submission · Major Revisions

It is still unclear how the candidate miRNAs were finally chosen for the validation experiments. The authors should find and report any potential miRNA involved in anthocyanin biosynthesis in potato. The story lacks the connection at certain points in the manuscript so the authors need to revise it appropriately.

Reviewer 1 ·

Basic reporting

This manuscript analyzed the miRNA in tubers of a purple tetraploid potato variety SD92 and its red mutant SD140. The English language should be improved to ensure that an international audience can clearly understand the text. Although there are some interesting results, the data analysis should be improved before it could be published.

Experimental design

The authors just list the results from sequence company without analysis

Validity of the findings

The raw data of the PCR need to be provided. Line 208-209, “One novel miRNA, novel_mir170, was down-regulated in SD140, while the corresponding target genes showed similar down regulations”, the miRNA and ist target gene have opposite expression pattern usually, how to explain it. In fact, the authors did not find any miRNA involving in anthocyanin biosynthesis in potato.

Additional comments

1. The introduction just list the references in this field. Line 64-65, “no miRNA analysis has been reported to be used in research of potato anthocyanin pathway”, please refer to the reference Front. Plant Sci., 11 December 2018 | https://doi.org/10.3389/fpls.2018.01742.
2. In the discussion, “Potato SD92 and SD140 may differ in biotic stress resistance”, is there any evidence?
3. Many errors in the references.
4. Statical analysis should be performed.

·

Basic reporting

The overall writing is good. The results and technical presentation are acceptable. Nonetheless, there are some confusing sentences, for example, in Line 166 & 168 "Both containing 77 genes", is it individually or collectively? Similar confusions have been found in many sentences. It is suggested to review the manuscript for such confusion.
Furthermore, There are a lot of studies on DEGs has been conducted in recent years. But there is no reference of advanced/recent research within 2021-2022 for similar studies in this draft.

Experimental design

research question and study are well designed.

Validity of the findings

Results are good and repeatable.

Additional comments

1: Line 124, How do the authors define the "low quality" of tags? It is suggested to add the selection criteria.
2: Different words have wrongly been capitalized for example line 155 "the members", Line 198, 201 ......"MiRNA".
3: Line 155: "significantly down-regulated" ,,, it is suggested to write the log2FC value to show "significance"
4: Line 163: "GO terms........" this full sentence" does not belong to results. it is suggested to delete it
5: Line 199: " target genes were....". how many? better to indicate with names and reason/criteria to select these genes for qPCR
6: Line 206: "However, its target,,,,,". the term "its" may need to replace with the article as "the" or proper name of miRNA.
7: Line 209: "corresponding target genes".... how many? which one? it is suggested to add the names of target genes here.
8: For the top candidate KEGG pathway, like the " plant harmon signaling pathway", it is suggested to add the pathway and highlight the candidate DEGs on it

Reviewer 3 ·

Basic reporting

Authors have tried to identify and validate the function of miRNAs involved in anthocyanin synthesis in Potato. They used two varieties of potato, a purple variety and a mutant variety. They prepared the RNA libraries of these varieties and performed different bioinformatic analysis to identify some known and novel putative miRNAs that could potentially play a role in anthocyanin synthesis.

Authors have provided all the data files including the raw files for the review. The manuscript is written well but a thorough grammatical and spell check is recommended.

Experimental design

Materials and methods need to described in little more detail. Some of the sections miss crucial information.

Validity of the findings

Authors have conducted different bioinformatic analyses to identify the putative miRNAs that could potentially be involved in anthocyanin biosynthesis. However, at some points, it is little unclear about how the candidate miRNAs were finally chosen for the validation experiments. Most of the results section mentions about different miRNAs and finally for the qRT-PCR different miRNAs were selected for the validation. Although it is understood that some of the miRNAs were already reported to be involved in anthocyanin, the expression data of those genes in this current study were not described appropriately in the results section. The story lacks the connection at certain points in the manuscript so the authors need to revise it appropriately. Authors should also discuss little more about the novel_miRNAs identified in their study and their possible role in anthocyanin biosynthesis.
Additional comments and suggestions are included in the edited version of the manuscript.

Annotated reviews are not available for download in order to protect the identity of reviewers who chose to remain anonymous.

---

## Round 0.2 · Minor Revisions

The reviewers are satisfied with the revised version. One of the Reviewers suggested the English language and grammar improvement. Please improve the grammar before its publication.

·

Basic reporting

The overall writing has been improved. The results and technical presentation are acceptable. The authors significantly improved the manuscript.

Experimental design

The research question and study were well-designed and executed.

Validity of the findings

Results are well-concluded and repeatable.

Additional comments

The authors have addressed the suggestions.

Reviewer 3 ·

Basic reporting

Authors have addressed the comments appropriately and have revised the manuscript. However, I still feel that there are some grammatical errors in the manuscript (particularly the tense). There are few minor corrections suggested in the edited version of the manuscript. Authors are advised to incorporate or address the suggestions made.

Experimental design

No comments

Validity of the findings

No comments

Additional comments

No comments

Annotated reviews are not available for download in order to protect the identity of reviewers who chose to remain anonymous.

---

## Round 0.3 · Minor Revisions

The authors have revised the manuscript by addressing all the queries. The manuscript is almost ready to be accepted for publication.

First, though, The Section Editor notes that:

> The manuscript needs editing for English/clarity.

> "Among them, 31 differentially expressed miRNAs were predicted to possess 305 target genes." miRNAs do not possess genes, do you mean potentially regulate or interact with these genes?

> "phenylalanine is a substrate," phenylalanine is not a substrate, it is an amino acid'

Please address these comments and have the manuscript edited before acceptance.

Reviewer 3 ·

Basic reporting

Authors have revised the manuscript by incorporating all the minor suggestions recommended. So the manuscript can be accepted.

Experimental design

No comments

Validity of the findings

No comments

---

## Round 0.4 · accepted · Accept

The authors have addressed the queries. The manuscript can now be accepted for publication.